# LAIR1, an ITIM-Containing Receptor Involved in Immune Disorders and in Hematological Neoplasms

**DOI:** 10.3390/ijms232416136

**Published:** 2022-12-17

**Authors:** François Van Laethem, Lucie Donaty, Emmanuelle Tchernonog, Vanessa Lacheretz-Szablewski, Jennifer Russello, Delphine Buthiau, Marion Almeras, Jérôme Moreaux, Caroline Bret

**Affiliations:** 1Department of Biological Hematology, CHU Montpellier, 34295 Montpellier, France; 2Department of Clinical Hematology, CHU Montpellier, 34295 Montpellier, France; 3Department of Biopathology, CHU Montpellier, 34295 Montpellier, France; 4Faculty of Medicine, University of Montpellier, 34090 Montpellier, France; 5CH Béziers, 34500 Béziers, France; 6Institute of Human Genetics, UMR 9002 CNRS-UM, 34396 Montpellier, France; 7Institut Universitaire de France (IUF), 75005 Paris, France

**Keywords:** LAIR1, autoimmunity, inflammation, inhibitory receptor, collagen, hematological neoplasms, immunoregulatory

## Abstract

Leukocyte-associated immunoglobulin (Ig)-like receptor 1 (LAIR1, CD305) belongs to the family of immune-inhibitory receptors and is widely expressed on hematopoietic mature cells, particularly on immune cells. Four different types of ligands of LAIR1 have been described, including collagens, suggesting a potential immune-regulatory function on the extracellular matrix. By modulating cytokine secretion and cellular functions, LAIR1 displays distinct patterns of expression among NK cell and T/B lymphocyte subsets during their differentiation and cellular activation and plays a major negative immunoregulatory role. Beyond its implications in physiology, the activity of LAIR1 can be inappropriately involved in various autoimmune or inflammatory disorders and has been implicated in cancer physiopathology, including hematological neoplasms. Its action as an inhibitory receptor can result in the dysregulation of immune cellular responses and in immune escape within the tumor microenvironment. Furthermore, when expressed by tumor cells, LAIR1 can modulate their proliferation or invasion properties, with contradictory pro- or anti-tumoral effects depending on tumor type. In this review, we will focus on its role in normal physiological conditions, as well as during pathological situations, including hematological malignancies. We will also discuss potential therapeutic strategies targeting LAIR1 for the treatment of various autoimmune diseases and cancer settings.

## 1. Introduction

Immune responses must be tightly regulated to prevent undesired events such as autoimmunity. A balance between cell activation and inhibition is required to modulate immune cellular events, relying on complex processes integrating signals generated from activating (AR) and inhibitory receptors (IR), these receptors being expressed by a large diversity of immune cells.

Leukocyte-associated immunoglobulin (Ig)-like receptor 1 (LAIR1), also termed CD305, is a type I glycoprotein of 287 amino acids, belonging to the family of immune IR, and was first molecularly cloned in 1997 [1].

Its structure contains one extracellular C2-type-Ig-like domain and two Immunoreceptor Tyrosine-Based Inhibitory Motif (ITIM) domains in its cytoplasmic tail. ITIMs are pivotal domains in the functionality of IR. They mediate negative control signals during the activation of immune cells, their effects being opposite to the action of Immunoreceptor Tyrosine-Based Activation Motifs (ITAMs), which are found in the intracellular domains of AR [2]. Phosphorylation of the tyrosine in the ITIM region is a central signaling event for the functionality of IR, including LAIR1, which require the phosphorylation of tyrosines of both ITIMs by Src family kinases in order to fully inhibit cellular responses [3]. ITIMs can then recruit and activate the Src homology 2 domain-containing tyrosine phosphatases SHP-1 and SHP-2, which in turn dephosphorylate cellular targets implicated in cell activation, to negatively regulate these processes [4].

Moreover, LAIR2, also known as CD306, is a soluble protein sharing 84% homology with LAIR1. LAIR2 can antagonize this homologous transmembrane receptor via competition for some ligands, regulating the inhibitory potential of LAIR1 [5]. Several splice variants of LAIR1 and LAIR2 have been identified, including LAIR1a, 1b, 1c, 1d, 2a, and 2b. Variations in structure have been described from one variant to another, but little is known about their functionality [6].

## 2. Expression Patterns of LAIR1

LAIR1 is widely expressed on almost all hematopoietic mature cells (Figure 1), including NK cells, T and B lymphocytes, monocytes [7], macrophages, basophils, mastocytes [8], eosinophils [9], dendritic cells [10], and a fraction of circulating plasmocytes [11].

LAIR1 is a collagen receptor expressed by the majority of immune cells, including T cells, B cells, NK cells, monocytes, neutrophils, macrophages, and pDCs, as well as tumor cells. Generally, LAIR1 acts as an immune-inhibitory receptor, which, upon ligand binding, blocks the activation and/or differentiation of these cells. The major effects of LAIR1 on each immune cell type are depicted.

In T lymphocytes, LAIR1 is heterogeneously expressed (Figure 2), the highest levels being described in CD8^+^ T cells compared to CD4^+^ T cells, and in naive populations in comparison with central or effector memory T cells in the CD4 and the CD8 compartments. These observations contrast with other IRs, known to be classically expressed after T cell activation. Moreover, in T cells, ligation of LAIR1 can inhibit T cell receptor (TCR)-mediated functions, suggesting that LAIR1 could contribute to the inhibition of the initiation of the immune response. On the contrary, TCR stimulation can upregulate LAIR1 expression on T cells, inhibiting effector functions [12].

In B lymphocytes, LAIR1 is expressed at high levels in naïve B cells, but only in 50% of memory B cells. Moreover, LAIR1 expression has been shown to be decreased upon cell activation [13].

In mature neutrophils, cell surface expression of LAIR1 is low or negative, as illustrated by Figure 2, this expression being progressively lost during granular differentiation. In addition, LAIR1 can be re-expressed at the surfaces of granulocytes after stimulation by tumor necrosis factor α, suggesting a regulatory role of this IR in the functionality of neutrophils [9].

**Figure 2 ijms-23-16136-f002:**
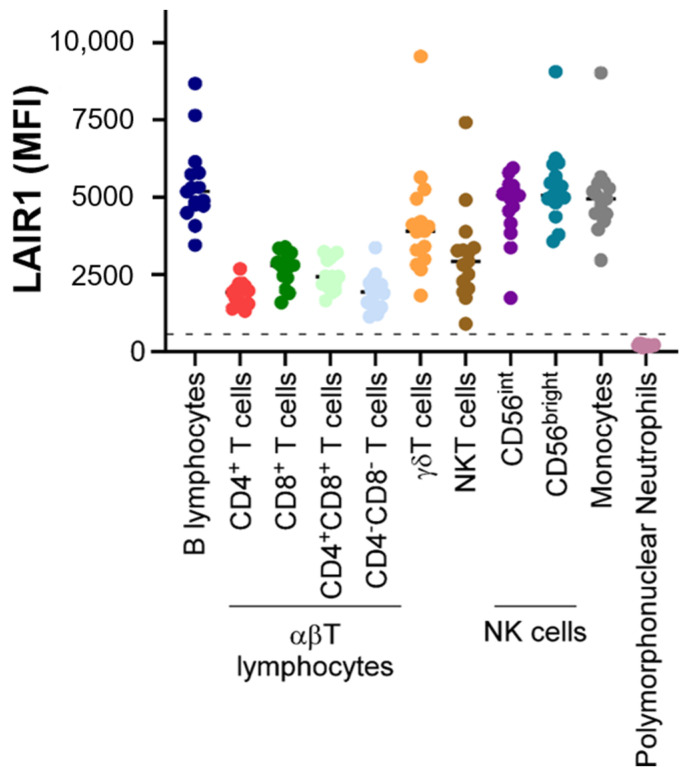
LAIR1 expression in normal white blood cells. LAIR1 expression was evaluated by flow cytometry (FCM) in a total of 10 samples obtained from healthy adult subjects (*n* = 10 blood samples, median age: 31 (range: 21–63); gender (male/female): 3/7) after red blood cell lysis by Versalyse© Beckman Coulter. For technical procedures, see [14]. Age (median): 31 (range: 21–63); gender (male/female): 3/7. Monocytes and granulocytes were selected based on their well-known CD45/side scatter (SSC) profile. In the “lymphocyte gate” determined on the graph CD45/SSC, T lymphocytes, NK cells, and B lymphocytes were selected according to the respective criteria of expression of CD3, absence of expression of CD3 and CD19 and expression of CD56, and expression of CD19 and CD20. The results obtained for the mean fluorescence intensity (MFI) of LAIR1 are presented. The dotted line represents the positivity threshold for this marker.

In macrophages, LAIR1 is highly expressed and can regulate their functions, including macrophage polarization [15]. In addition, LAIR1 can contribute to the inhibition of the differentiation of monocytes and dendritic cells from peripheral precursors [16].

Besides these mature cells, LAIR1 has been described as expressed by hematopoietic CD34^+^ progenitors [17] and by hematopoietic precursors as megakaryocytes, with a role as an early marker of differentiation and a negative regulator of thrombopoiesis, while not been detectable on platelets [18]. A murine model lacking LAIR1 was described as exhibiting peripheral thrombocytosis, increased proplatelet formation, a prolonged platelet half-life, enhanced Src kinase activity, and an increased risk of thrombus formation [19]. Thus, LAIR1 seems to be essential in the regulation of thrombopoiesis.

In non-hematopoietic cells, LAIR1 has been described to be expressed by osteoclasts, with a role in the inhibition of osteoclastogenesis [20].

## 3. LAIR1 Is a Receptor Able to Bind to Collagens and Complement Component 1Q, Surfactant Protein D, and Adiponectin

Four types of functional ligands have been described for LAIR1: collagens, complement component 1Q, surfactant protein D (SP-D), and adiponectin (ADP). Collagens are broadly expressed in vertebrates. In addition to their mechanical properties, they can be used as substrates for different cellular events, such as migration, and are involved in a large diversity of signaling pathways by binding to several receptors. Due to these properties, collagens display major roles in the maintenance of tissue structure and in blood coagulation. More recently, they have been described to be able to bind to LAIR1 with high affinity, particularly collagen I and III, via their conserved glycine–proline–hydroxyproline collagen repeat regions [21]. The identification of collagens as ligands for this IR highlights a potential immune-regulatory function for the extracellular matrix (ECM). In addition, LAIR1 could mediate cell adhesion to the ECM via the binding of collagens. Interestingly, LAIR2 can also bind collagens, which can lead to the efficient inhibition of the interaction between LAIR1 and collagens [6]. Moreover, collagens are supposed to be major ligands for LAIR1 in the hematopoietic system [22].

C1q, the first component of the complement, has been described to be able to contribute to complement-independent activities, besides its capacity for the initiation of the classic complement pathway. Among these activities, C1q, which contains collagen-like motifs, can bind to LAIR1, triggering the phosphorylation of ITIMs and leading to the modulation of immune cell differentiation and activation [23].

Surfactant protein D (SP-D) is a multimeric glycoprotein present at mucosal surfaces and involved in immune surveillance. SP-D exerts immunomodulatory activity via direct interactions with several receptors expressed by the epithelial cells of the mucosal tracts, as well as receptors expressed by innate and adaptive immune cells. SP-D contains collagen domains, allowing the binding of this glycoprotein to LAIR1, resulting in immunomodulatory effects [24].

More recently, it was demonstrated that adiponectin (APN), a key member of the adipokines mainly secreted by the white adipose tissue and able to modulate insulin resistance and glucidic and lipidic metabolism, could be a ligand for LAIR1, this interaction resulting in the inhibition of T cell activation [25].

## 4. An Inhibitory Receptor Involved in Immunity

Although first identified in human NK cells, the inhibitory function of this receptor has been extended to many different mononuclear cells. In human NK cells, crosslinking LAIR1 potently inhibits target cell lysis by both resting and activated cells in vitro [1]. Similarly, LAIR1 crosslinking inhibits B cell function [13], CD8+ T cell cytotoxicity [26], and TCR-induced cytokine production by T cell clones [27].

LAIR1 expression is also regulated in human and mouse T cells. Jansen and colleagues were the first to analyze its expression on human T cell subsets [28]. Not all mature T cells express LAIR1 as only 70–80% of CD4+ and between 80 and 90% of CD8+ T cells are LAIR1-positive. In addition, naïve CD4+ and CD8+ T cells, as well as effector CD8+ T cells, express higher levels of LAIR1 than memory cells. Stimulation of T cells through their T cell receptors (TCR) produced conflicting results. In one study, TCR crosslinking in vitro decreased LAIR1 expression on both CD4+ and CD8+ T cells at the protein and mRNA level [28]. In another study, TCR crosslinking increased the cell surface expression of LAIR1 on human T cells and required signaling by p38 and ERK MAP kinases [12]. This LAIR1 upregulation occurred on both naïve and memory T cells. Interestingly, both studies showed that LAIR1 crosslinking on T cells by specific antibodies or with natural ligands (collagens) inhibited T cell function, such as cytokine production or cytotoxic activity, after polyclonal or antigen-specific stimulation, revealing an important negative role of LAIR1 in T cell function [26]. In the absence of LAIR1 in vivo, only mild physiological effects were observed as mice deficient for LAIR1 showed normal T cell thymic lymphopoiesis and small decreases in peripheral splenic T cell numbers [29]. A small increase in peripheral T regulatory CD4+ T cells, as well as higher frequencies of activated and memory T cells, were also described. It was accompanied by a lower ability of CD4+ T cells in helping B cells with their optimal isotype class switching, although it is not clear if this was due to the absence of LAIR1 on B or on T cells. Only conditional knockdown mice for each individual subset will allow us to clarify this ambiguity. In conclusion, T cell function can clearly be inhibited by LAIR1 in vitro, but few noticeable effects can be noticed in its absence in vivo.

The expression of LAIR1 varies during different stages of B cell differentiation. The earliest stages in the bone marrow (BM), including pre-B cells and naïve B cells, express high levels of LAIR1 [13]. Only a portion of mature B cells in the bone marrow express LAIR1, and its expression is absent on BM-residing plasma cells. In peripheral blood and organs, naïve B cells show high LAIR1 expression and memory cells are divided into LAIR1-positive and -negative subsets. Interestingly, the high levels of LAIR1 on naïve cells are significantly diminished following B cell activation [30]. Concomitant with reduced intracellular Ca^2+^ release, stimulation of naïve B cells with a combination of CD40L/IL-4 and anti-LAIR1 monoclonal antibodies led to a significant decrease in IgG and IgE secretion, as well as reduced cytokine production (such as IL-10, IL-8, and TNF-a) [30]. As mentioned in the previous section on T cells, reduced class switching by activated B cells was also confirmed in vivo, using LAIR1-deficient mice [29]. In addition, LAIR1-deficient mice had only slight changes in B cell peripheral subsets, including a small increase in marginal zone B cells. In conclusion, LAIR1 seems to play a potential inhibitory role during B cell differentiation and activation.

NK cells’ cytotoxicity activity is well known to be regulated by numerous activating and inhibitory receptors. One of these relevant inhibitory receptors is LAIR1, which is expressed by the vast majority of NK cells (>90%). NK cell activation, through the CD16 receptor, is strongly inhibited following LAIR1 crosslinking in vitro, including the inhibition of target cell lysis or Ca^2+^ mobilization [1,26]. Interestingly, collagen was also shown to inhibit peripheral NK-mediated cytotoxicity and thereby contributes to the anti-inflammatory properties of the tumor microenvironment [31].

The expression of LAIR1 has been found to vary among the different circulating monocyte and dendritic cell subsets. Indeed, intermediate CD14^+^ monocytes express the highest levels of LAIR1, compared to the classical and non-classical monocyte subsets [32]. LAIR1 is also involved in the differentiation of monocytes towards dendritic cells. In vitro crosslinking of LAIR1 on peripheral blood mononuclear cells with monoclonal antibodies restrained their differentiation into dendritic cells in the presence of GM-CSF [16]. Recently, the ligation of LAIR1 by natural ligands such as C1q and C1 complexes was also able to inhibit the differentiation of monocytes into dendritic cells, including IFN-alpha production [23]. Different in vitro and in vivo models have shown that inflammation stimuli are capable of upregulating LAIR1 expression on monocytes [32,33,34]. Interesting interplays between LAIR1 expression on monocytes and T–B cell interactions have recently been uncovered. It was indeed shown by Meylaard and colleagues that LAIR1 can regulate the expression of the costimulatory molecule CD80 and HLA-DR, as well as modulating the LPS-TLR4 and IFN-alpha-mediated responses in human monocytes [32]. In conclusion, LAIR1 is expressed during inflammatory conditions through various stimuli and is capable, through its effects on monocytes and dendritic cells, of fine-tuning immune responses as well as restricting their differentiation.

Our antimicrobial defense mechanisms strongly rely on the presence of neutrophils, but their inappropriate accumulation leads to inflammation and immune pathology. There is some evidence that LAIR1 is differentially expressed during their differentiation and may be involved in their function. Indeed, bone-marrow-derived neutrophil precursors express high levels of LAIR1, but more differentiated cells have reduced surface levels. Although circulating neutrophils do not express significantly LAIR1, its upregulation can rapidly be induced following in vitro activation with cytokines [35,36]. Functionally, the analysis of LAIR1-deficient mice has shown that LAIR1 controls airway neutrophil responses in vivo. Upon challenge with a variety of respiratory stimuli, such as respiratory syncytial virus or cigarette smoke, LAIR1-deficient mice showed increased airway inflammation accompanied by increased lymphocyte and neutrophil airway recruitment, but with no apparent consequence for viral loads or cytokine production [37]. In the case of cigarette smoke exposure, the absence of LAIR1 also enhanced neutrophil recruitment to the airways and worsened the severity of the disease. An intrinsic function for LAIR1 on neutrophils was validated when intranasal chemokine-induced neutrophil recruitment was increased in the airways in LAIR1-deficient mice [37]. To conclude, LAIR1 can suppress neutrophil tissue migration in lung infection/disease and can negatively regulate neutrophil-driven airway inflammation. Therefore, LAIR1 could represent a promising target in airway-related diseases.

## 5. LAIR1 in Autoimmunity and Inflammatory Diseases

Protection against self-damage due to the loss of self-tolerance, hyperimmune activation, or autoimmunity relies on immune-inhibitory receptors expressed by both innate and adaptive immune cells. It is therefore not surprising that LAIR1 has a critical role in autoimmunity, given its widespread pattern of expression and its ability to suppress cell activation in a number of hematopoietic cells. It is thought that the inhibitory signals created through LAIR1 provide a threshold for immune activation, especially in collagen-rich tissues, thereby preventing uncontrolled immune responses and allowing the maintenance of immune cell homeostasis. In this section, we will discuss some instances in which LAIR1 has been shown to be involved in autoimmunity and inflammatory diseases.

## 6. LAIR1 and Allergy

To address its role in allergic responses, Omiya and colleagues generated a transgenic mouse system expressing a chimeric version of LAIR1. This chimeric protein contained its extracellular domain fused with an immunoglobulin tag and acted as a decoy by competing with endogenous LAIR1 [38]. Using a murine experimental model of allergic dermatitis, the authors showed that the decoy LAIR1 transgenic mice had an increased susceptibility to contact hypersensibility (CHS). This increased susceptibility to CHS happened both at the sensitization and elicitation phases and implicated the repression of cytokine production (IL-6 and IL-12) by dendritic cells and the inhibition of proliferation and cytokine of both naïve and memory T cells.

Similarly, the role of LAIR1 in airway hyper-reactivity and type 2 asthma was recently addressed by Helou and colleagues [39]. They were able to demonstrate that LAIR1 can be induced on activated pulmonary ILC2, leading to decreased cytokine secretion and effector functions. Airway hyper-reactivity was indeed exacerbated in the absence of LAIR1 in both mouse and human models, highlighting its normal protective anti-inflammatory role. Interestingly, circulating human ILC2 shows surface LAIR1 expression, which can be further increased upon IL-33 stimulation. This contrasts with mouse ILC2, which only expresses LAIR1 upon cytokine stimulation [39]. It still remains to be seen how LAIR1 can be targeted clinically for therapeutic purposes in the case of asthma.

## 7. LAIR1 and Lupus

The complement system is involved in both innate and adaptive immune systems and has been shown to be important in the pathogenesis of systemic lupus erythematosus (SLE). Deficiencies in some members of the classical pathway, such as C1q, C4, and C2, can predispose individuals to the development of autoimmune diseases, explained in part by their important roles in promoting the clearance of immune complexes and apoptotic cells [40]. Furthermore, C1q has been shown to regulate, via LAIR1, myeloid cell activity and prevent the unwarranted activation of circulating monocytes and the development of mono-derived dendritic cells [23,41]. At the protein level, the collagen tail of C1q interacts with LAIR1, whereas C1q’s globular head interacts with the transmembrane receptor CD33 (Siglec-3), inducing the phosphorylation of the ITIM motifs in the tail of CD33, and altogether forms a C1q/CD33/LAIR1 inhibitory ternary complex [42]. Disruption of this inhibitory complex is frequently found in SLE, consistent with abnormalities related to C1q or a lack of LAIR1 and CD33 expression on circulating SLE myelomonocytes observed in SLE patients [42,43]. Aberrant C1q function may be linked to impaired LAIR1 expression and may be CD33-independent, since LAIR1 has been shown to be abnormally expressed on B cells and plasmacytoid DCs (pDCs) in SLE patients [44]. Defective LAIR1 expression in both B cells and pDCs, as well as the subsequent decreased inhibitory signals, lead to abnormal cell activation and may contribute to SLE development, highlighting the critical role of complement-mediated inhibition through LAIR1 in controlling immune homeostasis in normal and anomalous physiology.

## 8. LAIR1 and Rheumatoid Arthritis

Rheumatoid arthritis (RA) is a common autoimmune disease associated with progressive disability and systemic complications, characterized by events such as synovial inflammation, autoantibody production, and cartilage and bone destruction [45]. Patients with RA have lower surface expression of LAIR1 on their fibroblast-like synoviocytes (FLS) but significantly higher levels of soluble LAIR1 and its secreted homolog LAIR2 in the serum and synovial fluid [46]. Its expression is used as a biomarker when analyzing synovial fluid for evidence of RA [34].

Although FLS expressed low levels of surface LAIR1, treatment of these cells with TNF-alpha induced LAIR1′s shedding. A reduction in FLS invasion could be achieved by LAIR1 overexpression, as well as a decrease in inflammatory cytokines such as IL-6 and IL-8. All these data suggest that LAIR1 serves as an anti-inflammatory molecule in RA. LAIR1 expression is also affected on T cells from RA patients. As similarly observed with FLC cells, reduced surface LAIR1 expression on CD4+ T cells was seen in RA patients, leading to the decreased inhibition of T cell activation and hyperactivation. Mice rendered deficient for LAIR1 show more severe arthritis than their wildtype counterparts [47]. The mechanisms responsible for the inhibition of TCR signaling by LAIR1 engagement have recently been addressed by Park and colleagues. They were able to show that LAIR1 stimulation by collagen inhibits TCR signaling by decreasing the phosphorylation of key components of the TCR signaling pathway, such as Lck, ZAP-70, c-Jun, and p38 [48]. The kinase Csk, known to bind the intracellular tail of LAIR1 after phosphorylation of the ITIM motifs [1,36], was shown to be essential for the LAIR1-dependent inhibition of the TCR signaling cascade [48]. In conclusion, LAIR1 engagement by collagen-like domains could be an interesting therapeutic strategy to control inflammation in autoimmune diseases such as RA, SLE, and many other inflammatory states. Interestingly, it was recently observed that the effect of the vitamin D could be at least partially mediated by the upregulation of LAIR1 in a model of arthritis [49].

## 9. LAIR1 and Graft Rejection

The control of immune activation is essential during cell or organ transplantation, and it is therefore crucial to better understand the role played by key negative regulators of lymphocyte activation in such conditions. Recently, the role of LAIR1 has been addressed in a murine chronic lung model, as well as heart allograft rejection models [50]. The authors observed a significant decrease in cellular infiltration in the heart grafts from LAIR1-deficient mice as compared with wildtype mice, as well as a lower amount of collagen deposition and vascular rejection. Similar results were also observed in a lung transplantation model [50]. In a more clinically relevant study, it was shown that levels of soluble LAIR1 were elevated in the sera of transplant rejection patients compared to non-rejection patients and healthy individuals [51], showing the clinical relevance of the increased shedding of membrane-bound LAIR1. Both human and mouse studies suggest that LAIR1 could be an important regulatory target in organ transplantation settings and its soluble form could be used as a useful predictor of lymphocyte activation.

## 10. LAIR1 and Liver Cirrhosis

The expression of LAIR1 was studied on liver macrophages and blood monocytes obtained from patients having progressive liver disease. LAIR1 was indeed expressed by hepatic macrophages, but the number of positive cells was reduced in cirrhotic patients. On peripheral monocytes, higher levels of LAIR1 were found in cirrhotic patients, even at early clinical stages, compared to healthy individuals [52]. A reduced frequency of LAIR1-expressing T cells was also found in biopsies of patients with liver fibrosis [53]. Increased collagen deposition is a feature of liver disease and this makes its receptor LAIR1 more relevant. All these results pinpoint that LAIR1 could be considered a promising biomarker in the diagnosis and evaluation of hepatic cirrhosis disease progression.

## 11. A Potential Contradictory Role in Tumor Biology

LAIR1 has been largely described as involved in the regulation of immune cellular processes. Its implication in T cell immunosuppression has been documented in the context of the tumor microenvironment as a means to evade anti-cancer immune responses [54]. In the example of lung tumor models, LAIR1 was able to promote collagen-induced CD8^+^ T cell exhaustion [55].

Other recent observations are in favor of the implication of this receptor in neoplastic diseases when directly expressed by tumor cells. In solid tumors, the overexpression of LAIR1 has been shown on the surfaces of cancer cells in epithelial ovarian cancer [56], cervical cancer [57], breast cancer [58], oral squamous cell carcinoma (OSCC) [59], hepatocellular carcinoma [60], and renal cell carcinoma (RCC) [61], whereas the expression of this IR was not detected in normal equivalent tissue. In addition, from a clinical point of view, the level of expression of LAIR1 has been correlated with tumor aggressiveness (Table 1). In particular, in invasive breast cancer, high LAIR1 expression was associated with a higher histological grade and a poor clinical outcome. It was also associated with the biological characteristics of cancer cells linked to treatment response, such as hormone receptor negativity, and with physio-pathological aspects such as immune cell infiltrates and extracellular matrix remodeling events [58]. In OSCC, LAIR1 was described as significantly overexpressed in the stroma zone in comparison with adjacent normal mucosa and dysplasia areas. In addition, the overexpression of LAIR1 was associated with advanced pathological grades and immune-suppressive profiles [59]. In hepatocellular carcinoma, a high level of expression of LAIR1 in cancer tissue was associated with poor cancer differentiation and with worse overall survival [60]. In RCC tumor tissue, LAIR1 was observed as significantly upregulated compared to normal renal tissue. The high expression of LAIR1 in this tissue was correlated with poor progression-free survival [61]. In osteosarcoma, an increase in LAIR1 expression was associated with an advanced stage [62].

The opposite effects of LAIR1 were observed in vitro depending on the cancer cell type (Table 2). In the ovarian cancer cell line HO8910, inhibition of LAIR1 expression was shown to promote in vitro cell proliferation, to increase clonogenicity during colony formation assays, and to favor the invasive properties of tumor cells [56]. In the cervical cancer cell line ME-180, the overexpression of LAIR1 was described to inhibit their proliferation and to reverse their anti-apoptosis tendency [57]. These observations are in accordance with the anti-tumoral properties mediated by LAIR1.

On the contrary, the knockdown of LAIR1 in the BC cell lines SKBr3 and MDA-MB 231 reduced cell proliferation and impaired cell invasion properties. The high expression of LAIR1 at the mRNA level in clinical samples was positively correlated with collagens, suggesting that the high expression of collagens within the tumor microenvironment may lead to pro-tumoral responses via LAIR1 [58]. In the Caki-1 and Caki-2 RCC human cell lines, the knockdown of LAIR1 induced the decreased phosphorylation of Akt, the PI3K/Akt/mTOR pathway being a major axis for oncogenic events in RCC, including proliferation. The opposite effect was observed when LAIR1 was overexpressed in the RCC human cell line ACHN, the result being the upregulation of the level of Akt phosphorylation [61].

In hematological malignancies, several studies have examined the potential implication of LAIR1 in tumorigenic processes. In myelomonocytic leukemic human cell lines, engagement of LAIR1 was described to induce cell cycle arrest in the G_0_/G_1_ phase, resulting in the inhibition of proliferation and in apoptosis [63].

The same effects were observed for primary acute myeloid leukemia (AML) blasts isolated from peripheral blood or from bone marrow samples, cultured in vitro and stimulated by granulocyte–monocyte colony-stimulating factor (GM-CSF). Of note, LAIR1 can potentially be expressed by human primary blastic cells from the M0 to M5 stages of the old French American and British (FAB) classification [64]. In contrast, high levels of LAIR1 mRNA in primary AML blasts were correlated with poor overall survival in a cohort of 104 AML patients younger than 65 years old. The knockdown of LAIR1 in human leukemia cell lines resulted in an increase in apoptosis and in the suppression of in vitro cellular growth, indicating that LAIR1 could be essential to support AML development. These observations were reinforced by the fact that AML blastic cells deficient in LAIR1 failed to engraft in a murine model. In addition, LAIR1 promoted the activity of AML stem cells [22].

In Philadelphia chromosome positive B acute lymphoblastic leukemia (Ph^+^ B ALL), LAIR1 was observed to be highly expressed on primary blastic cells. Higher expression of this IR was correlated with shorter overall and relapse-free survival. In vitro, ITIM cytoplasmic motifs of LAIR1 were described as critical for the survival of B ALL cells [65].

**Table 2 ijms-23-16136-t002:** Opposite effects of LAIR1 observed in vitro in several cancer cell lines (↑: increase; ↓: decrease).

In Vitro Anti-Tumoral Properties of LAIR1	In Vitro Pro-Tumoral Properties of LAIR1
**Ovarian cancer:** Inhibition of LAIR1 expression in HO8910 cell line: ↑ proliferation, clonogenicity, and invasive properties of tumor cells [56]	**Breast cancer:** Knockdown of LAIR1 in SKBr3 and MDA-MB 231 cell lines: ↓ proliferation and impaired cell invasion properties [58]
**Cervical cancer:** Overexpression of LAIR1 in ME-180 cell line: inhibition of proliferation and reversion of the anti-apoptosis tendency [57]	**Renal cell carcinoma:** Knockdown of LAIR1 in Caki-1 and Caki-2 cell lines: ↓ Akt phosphorylation [61]
	**Acute Myeloid Leukemia (AML):** Knockdown of LAIR1 in AML cell lines: ↓ cell growth and ↑ apoptosis; failure to engraft in a murine model [22]
	**B Acute Lymphoblastic Leukemia (B ALL):** ITIM cytoplasmic motifs of LAIR1 described as critical for the survival of B ALL cells [65]

In a study performed by FCM on malignant cells obtained from a large group of samples obtained from tissular or circulating phases of B cell chronic lymphoproliferative diseases (B-CLPD), we observed that this receptor was significantly downregulated (negative to low expression in comparison with normal B peripheral cells) by tumoral cells from Burkitt Lymphoma (BL), High-Grade B Cell Lymphoma (HGBCL), Diffuse Large B Cell Lymphoma (DLBCL), Follicular Lymphoma (FL), and Waldenström Macroglobulinemia (WM). Of note, in rare cases of exception for HGBCL, DLBCL, and WM, we observed a level of expression of LAIR1 similar to that of normal B cells (Figure 3).

In malignant B cells from Chronic Lymphocytic Leukemia (CLL) and Splenic Marginal Zone Lymphoma (SMZL), LAIR1 was heterogeneously expressed (Figure 3) in comparison with the level of expression in normal B cells, with two dichotomic groups of patients defined as patients with B tumoral cells expressing LAIR1 and patients with B malignant cells that did not express LAIR1. In CLL, it is known that the level of expression of LAIR1 is associated with prognosis. First, it was observed that this receptor was expressed by CLL cells obtained from patients with low or intermediate risk, but it was absent at the surfaces of CLL cells obtained from high-risk patients [66]. In addition, higher expression of LAIR1 was observed on CLL cells in the context of monoclonal B cell lymphocytosis, compared with CLL cells with adverse chromosomal abnormalities [67]. In a more recent study, it was shown that the expression of LAIR1 in CLL cells from a cohort of 311 patients was inversely correlated with the level of CD38 expression. A more advanced Binet stage, the presence of high-risk cytogenetic abnormalities, and an unmutated immunoglobulin heavy-chain genetic profile were associated with lower levels of LAIR1 expression [68].

In contrast with these profiles and in concordance with our results (Figure 3), this receptor was significantly upregulated in malignant cells from Hairy Cell Leukemia (HCL), this marker being introduced in FCM B Euroflow consensus panels to identify this B-CLPD entity [69]. To date, more studies are required to understand the precise role of LAIR1 in the different entities of B-CLPD.

## 12. High LAIR1 Expression Is Associated with a Poor Outcome in Several Hematological Malignancies

Using the Genomicscape webtool [70], we investigated the prognostic value of *LAIR1* expression in different hematological malignancies. Taking advantage of publicly available cohorts of newly diagnosed patients, the prognostic value of *LAIR1* expression was investigated in AML with a normal karyotype (Gene Expression Omnibus (http://www.ncbi.nlm.nih.gov/geo/ (accessed on 8 October 2022)) under accession number GSE12417 “GSE12417 - Prognostic gene signature for normal karyotype AML - OmicsDI (accessed on 17 February 2022), *n* = 78 [71,72]), FL (GSE16131 “GSE16131 - Differences Between Follicular Lymphoma With and Without Translocation t(14;18) - OmicsDI (accessed on 17 February 2022), *n* = 180 [73]), MCL (GSE10793 “GEO Accession viewer (nih.gov) (accessed on 17 February 2022), *n* = 62 [74]), DLBCL (GSE10846 “GEO Accession viewer (nih.gov) (accessed on 17 February 2022), CHOP cohort *n* = 181 and R-CHOP cohort *n* = 233 [75,76]), and Multiple Myeloma (MM, GSE2658 “GEO Accession viewer (nih.gov) (accessed on 17 February 2022), *n* = 345 [77,78]), using the Maxstat R algorithm [79]. Of interest, high *LAIR1* expression demonstrated significant prognostic value in AML (*p* = 0.0006), DLBCL patients (*p* =0.01 and *p* = 0.003), and MM patients (*p* = 0.006) (Figure 4). These observations underline the potential role of LAIR1 in AML, DLBCL, and MM pathophysiology.

## 13. LAIR1 as a Future Therapeutic Target

Since LAIR1 is widely expressed on both innate and adaptive immune cells, it constitutes, therefore, a very interesting pharmaceutical target for a number of inflammatory and cancerous conditions. Moreover, and as mentioned above, collagen-rich environments can be found in many tumors and a higher density correlates with a poor prognosis [80,81]. This leads to poor lymphocyte recruitment and the direct stimulation of tumor growth. Thus, the disruption of collagen–LAIR1 interactions represents an important pharmaceutical strategy. LAIR2 is a soluble homolog of LAIR1 with more than 80% identity in the extracellular region and binds to collagens with higher affinity than LAIR1. LAIR2 serves as a natural agonist that blocks LAIR1-inhibitory signaling [5]. Meyaard and colleagues recently described the use of a dimeric LAIR2 Fc fusion protein called NC410 to target collagens in tumors and reverse immune suppression [82]. They were able to show that this fusion protein boosted T cell activity and had a therapeutic effect in T-cell-dependent tumor models. More experiments are needed to validate its potential clinical use as a single therapy or in association with other checkpoint inhibitors.

## 14. Conclusions

The inhibitory role of LAIR1 in immune regulation has been well documented, but its function during tumorigenesis and tumor progression remains contradictory. In vitro crosslinking of LAIR1 with monoclonal antibodies or with collagen provides negative inhibitory signals on a wide variety of immune cells, such as monocytes/dendritic cells, NK cells, as well as T and B lymphocytes. Its broad expression pattern, as well as the variety and abundance of its ligands, further emphasizes its crucial inhibitory and regulatory function. The presence of the soluble forms of LAIR1 and LAIR2 proteins, its expression levels, as well as the strength of the activating signals, are other factors that further regulate the degree of inhibition exerted by this receptor. In various in vivo inflammatory conditions, deregulation of LAIR1 function also leads to increased and uncontrolled inflammation, with potential harmful consequences. Likewise, its upregulation in some types of cancer has been associated with disease progression and can act as a predictor of clinical outcomes. Furthermore, stimulating LAIR1 can affect tumor growth. In conclusion, there is clear evidence that LAIR1 represents a very attractive therapeutic target for the treatment of a variety of inflammatory and autoimmune diseases, as well as hematological malignancies.

## Figures and Tables

**Figure 1 ijms-23-16136-f001:**
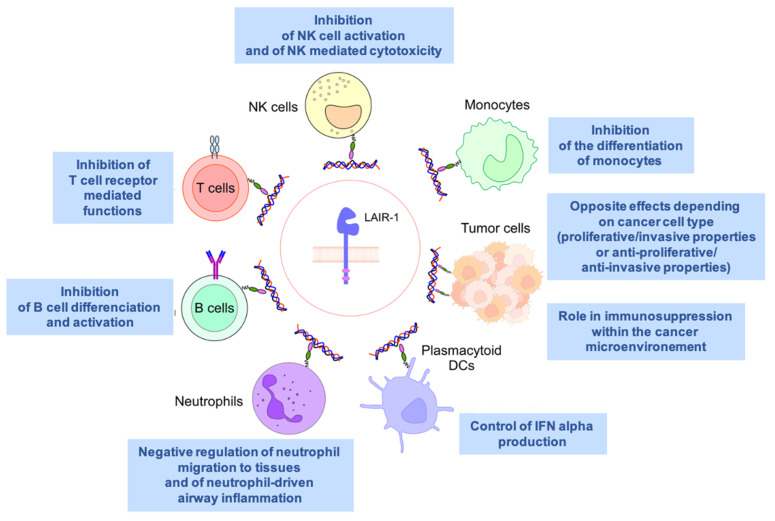
Summary of LAIR1 expression and functions in normal leucocytes, plasmacytoid dendritic (pDCs) cells, and tumor cells.

**Figure 3 ijms-23-16136-f003:**
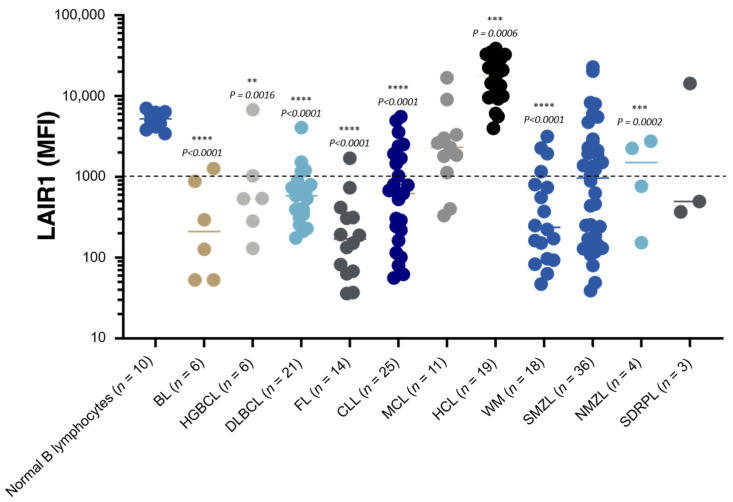
LAIR1 expression in B cell lymphoma and comparison with normal B cells. LAIR1 expression was evaluated by flow cytometry (FCM) in a total of 163 samples obtained from patients with B cell chronic lymphoproliferative diseases at diagnosis, including *n* = 6 samples from patients with Burkitt Lymphoma (BL), *n* = 6 samples from patients with High-Grade B Cell Lymphoma (HGBCL), *n* = 21 samples from patients with Diffuse Large B Cell Lymphoma (DLBCL), *n* = 14 samples from patients with Follicular Lymphoma (FL), *n* = 25 samples from patients with Chronic Lymphocytic Leukemia (CLL), *n* = 11 samples from patients with Mantle Cell Lymphoma (MCL), *n* = 19 samples from patients with Hairy Cell Leukemia (HCL), *n* = 18 samples from patients with Waldenström Macroglobulinemia (WM), *n* = 36 samples from patients with Splenic Marginal Zone Lymphoma (SMZL), *n* = 4 samples from patients with Nodal Marginal Zone Lymphoma (NMZL), and *n* = 3 samples from patients with Splenic Diffuse Red Pulp Lymphoma (SDRPL). Median age: 70 and range: 8–93; gender (male/female): 102/61 (*n* = 99 blood samples, *n* = 50 bone marrow aspiration, *n* = 6 lymph node/splenic biopsies/cerebral biopsies, *n* = 8 serous liquid). In parallel, the level of expression of LAIR1 by normal B cells was evaluated by FCM in a total of 10 samples obtained from healthy adult subjects as described in Figure 2. Boxplots illustrate the mean fluorescence intensity (MFI) of LAIR1 on a logarithmic scale. The dotted line represents the positivity threshold for this marker. For technical procedures, see [14]. Statistical differences were determined between MFI of normal B cells and MFI of each type of B lymphoma using Prism 9© (unpaired *t*-test). Only significant results are presented (****, ***, or **).

**Figure 4 ijms-23-16136-f004:**
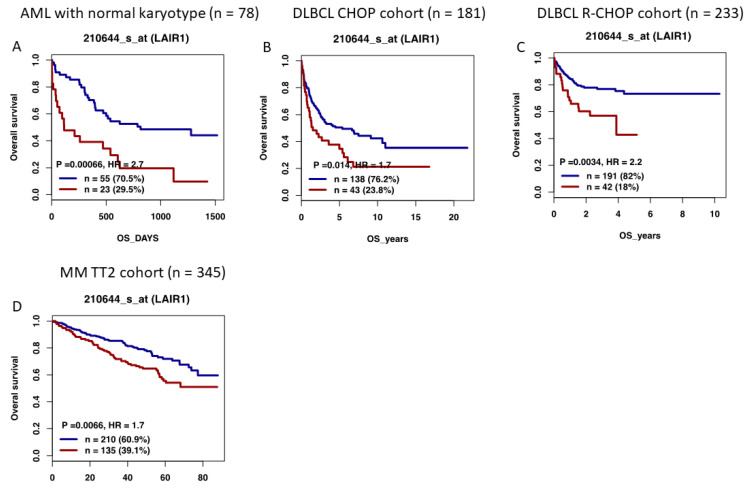
Prognostic value of LAIR1 expression in patients with DLBCL. (**A**) Cytogenetically normal AML patients from the GSE12417 cohort (*n* = 78) were ranked according to increased LAIR1 expression (Affymetrix U133P microarrays), and the maximum difference in OS was obtained by splitting patients into high-risk and low-risk groups. (**B**,**C**) High LAIR1 expression also had prognostic value in DLBCL, including the Lenz CHOP cohort *(n* = 181) and Lenz R-CHOP cohort (*n* = 233) (Affymetrix U133P microarrays). (**D**) LAIR1 expression was also significantly associated with a poor outcome in MM (n = 345, Affymetrix U133P microarrays). Blue: low expression of LAIR1; red: high expression of LAIR1.

**Table 1 ijms-23-16136-t001:** Solid cancer models with clinical observations linking higher expression of LAIR1 to cancer aggressiveness.

**Breast cancer**	Association of high expression of LAIR1 with higher histological grade and poor outcome, hormone receptor negativity, extracellular matrix remodeling events [58]
**Oral squamous cell carcinoma**	Association of high expression of LAIR1 with advanced pathological grades and immune-suppressive profiles [59]
**Hepatocellular carcinoma**	Association of high expression of LAIR1 with poor cancer differentiation and with worse overall survival [60]
**Renal cell carcinoma**	Association of high expression of LAIR1 with poor progression-free survival [61]
**Osteosarcoma**	Association of high expression of LAIR1 with advanced stage [62]

## Data Availability

Not applicable.

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
