# Peer review of "LAIR1, an ITIM-Containing Receptor Involved in Immune Disorders and in Hematological Neoplasms"

_ijms, 2022, doi:10.3390/ijms232416136_

Round 1
Reviewer 1 Report
this is a very well written review. It can be published without modifications. congratulations to authors.
Author Response
We thank the reviewer for this very positive evaluation of our work.
Reviewer 2 Report
The theme of the article is clear, which is instructive to medical work and worth publishing
Author Response
We thank the reviewer for this very positive comment on our work.
Reviewer 3 Report
Outcomes interesting because in malignant B cells LAIR1 was expressed heterogeneously in comparison with normal cells B.
Author Response
We thank the reviewer for this very positive comment on our work. Indeed, the level of expression of LAIR1 at the surface of B lymphoma cells was only partially described in the literature. For this reason, we have integrated in our article these original data, obtained from a large series of samples of B lymphoma entities.
Reviewer 4 Report
In this review authors discuss the role of LAIR1 in pathological and normal physiological conditions as well as its potential as a therapeutic target in auto-immune diseases and cancer Authors give a detailed introduction about LAIR1, explained general features of LAIR1 protein and outcomes of LAIR1 expression differences in multiple conditions.
Some minor points to address:
Line 55. “Several splice variants of LAIR1 have been identified, including LAIR1a, 1b, 1c, 2a and 2b.” Are there any differences between these splice variants, such as functionality?
Figure 1. Arrows in the figure with different directions are confusing. Please indicate what these arrows mean in the figure legend.
Line 77. “Moreover, in T cells, ligation of LAIR1 can inhibit T cell receptor (TCR) mediated functions, suggesting that LAIR1 could contribute to the inhibition of the initiation of immune response.” What is the mechanism of action of LAIR1 to inhibit TCR mediated functions?
Line 110. “In non-hematopoietic cells, LAIR1 has been described to be expressed by osteoclasts”. Is the function of LAIR1 known in osteoclasts?
Since LAIR1 has different types of functions in different conditions, it can be difficult to follow. Thus, it would be a good summary if authors provide a table for the readers to follow and understand this review.
Author Response
- Line 55. “Several splice variants of LAIR1 have been identified, including LAIR1a, 1b, 1c, 2a and 2b.” Are there any differences between these splice variants, such as functionality?
We thank the reviewer for this question. This point has been discussed by Line Meyaard in a review published in 2008 in the Journal of Leukocyte Biology. Variations in structure have been described from one variant of LAIR-1 and LAIR-2 to another, but little is known about their functionality. We added this information in the reviewed manuscript (line 57).
In comparison with the full-length form of LAIR-1a, a series of 17 amino acids is missing in the region located between the transmembrane domain and the Ig like domain of the LAIR-1b form. The same observation was done for LAIR-2b, in comparison with the full-length LAIR-2a form. These modifications may affect their level of glycosylation. A differential expression of LAIR-1a and LAIR-1b has been hypothesized in NK and T cells, but need to be confirmed.
LAIR-1c shares the same structure as LAIR-1b, with the exception of a single amino acid change in the extracellular domain. Concerning LAIR-1d, a part of the intracellular domain of the molecule is truncated.
- Figure 1. Arrows in the figure with different directions are confusing. Please indicate what these arrows mean in the figure legend.
We thank the reviewer for the comment. For greater clarity, we have removed the arrows and replaced them with text labels in a new version of this Figure 1 included in the revised manuscript.
- Line 77. “Moreover, in T cells, ligation of LAIR1 can inhibit T cell receptor (TCR) mediated functions, suggesting that LAIR1 could contribute to the inhibition of the initiation of immune response.” What is the mechanism of action of LAIR1 to inhibit TCR mediated functions?
The mechanism is described in the paragraph dedicated to the role of LAIR-1 during rheumatoid arthritis (from line 310 to line 317): “The mechanisms responsible for the inhibition of TCR signaling by LAIR1 engagement have recently been addressed by Park and colleagues. They were able to show that LAIR1 stimulation by collagen inhibits TCR signaling by decreasing the phosphorylation of key components of the TCR signaling pathway such as Lck, ZAP-70, c-Jun and p38 [48]. The kinase Csk, known to bind the intracellular tail of LAIR1 after phosphorylation of the ITIM motifs [1,36], was shown to be essential for the LAIR1 dependent inhibition of the TCR signaling cascade [48]. »
- Line 110. “In non-hematopoietic cells, LAIR1 has been described to be expressed by osteoclasts”. Is the function of LAIR1 known in osteoclasts?
In their article published in 2013, Zhang and colleagues have shown that LAIR-1 inhibits osteoclastogenesis. We have added this information in the revised manuscript (line 123).
- Since LAIR1 has different types of functions in different conditions, it can be difficult to follow. Thus, it would be a good summary if authors provide a table for the readers to follow and understand this review.
We thank the reviewer for this suggestion. In addition to the general information provided in Figure 1, we have added 2 tables summarizing the effects of LAIR-1 in the different cancer models for which there is indeed a great diversity of effects depending on the tumor type:
- Table 1, mentioned line 358 and integrated at line 378:
Breast cancer |
Association of a high expression of LAIR1 with higher histological grade and poor outcome, hormone receptor negativity, extracellular matrix remodeling events [58] |
Oral squamous cell carcinoma |
Association of a high expression of LAIR1 with advanced pathological grades and immune suppressive profiles [59] |
Hepatocellular carcinoma |
Association of a high expression of LAIR1 with a poor cancer differentiation and with a worse overall survival [60] |
Renal cell carcinoma |
Association of a high expression of LAIR1 to a poor progression-free survival [61] |
Osteosarcoma |
Association of a high expression of LAIR1 with advanced stage [62] |
Table 1. Solid cancer models with clinical observations linking a higher expression of LAIR1 to cancer aggressiveness
- and table 2, mentioned line 380 and integrated at line 419:
In vitro ANTITUMORAL properties of LAIR1 |
In vitro PROTUMORAL properties of LAIR1 |
Ovarian cancer: Inhibition of LAIR1 expression in HO8910 cell line: augmentation of proliferation, clonogenicity and invasive properties of tumor cells [56]
|
Breast cancer: Knockdown of LAIR1 in SKBr3 and MDA-MB 231 cell lines: diminution of proliferation and impaired cell invasion properties [58] |
Cervical cancer: Overexpression of LAIR1 in ME-180 cell line: inhibition of proliferation and reversion of the anti-apoptosis tendency [57]
|
Renal cell carcinoma: Knockdown of LAIR1 in Caki-1 and Caki-2 cell lines: diminution of Akt phosphorylation [61] |
Acute Myeloid Leukemia (AML): Knockdown of LAIR1 in AML cell lines: diminution of cell growth and augmentation of apoptosis; failure to engraft in a murine model [22] |
|
B Acute Lymphoblastic Leukemia (B ALL): ITIM cytoplasmic motifs of LAIR1 described as critical for the survival of B ALL cells [65] |
Table 2. Opposite effects of LAIR1 observed in vitro in several cancer cell lines